# SARS-CoV-2 Neutralizing Antibodies in Mexican Population: A Five Vaccine Comparison

**DOI:** 10.3390/diagnostics13061194

**Published:** 2023-03-22

**Authors:** Fernando Alcorta-Nuñez, Diana Cristina Pérez-Ibave, Carlos Horacio Burciaga-Flores, Miguel Ángel Garza, Moisés González-Escamilla, Patricia Rodríguez-Niño, Juan Francisco González-Guerrero, Adelina Alcorta-Garza, Oscar Vidal-Gutiérrez, Genaro A. Ramírez-Correa, María Lourdes Garza-Rodríguez

**Affiliations:** 1Servicio de Oncología, Centro Universitario Contra el Cáncer (CUCC), Hospital Universitario “Dr. José Eleuterio González, Universidad Autónoma de Nuevo León, Monterrey 64460, Nuevo León, Mexico; fernando.alcortannz@uanl.edu.mx (F.A.-N.); dperezi@uanl.edu.mx (D.C.P.-I.); carlos.hburciaga@gmail.com (C.H.B.-F.); moisesgzz87@gmail.com (M.G.-E.); patriciardzn@hotmail.com (P.R.-N.); juanfglz@hotmail.com (J.F.G.-G.); adelina.alcortagr@uanl.edu.mx (A.A.-G.); vidal_oscar@hotmail.com (O.V.-G.); 2ONCARE Treatment Center, Valle Unit, San Pedro Garza García 66220, Nuevo León, Mexico; 3Unidad Médica de Alta Especialidad, Hospital de Gineco Obstetricia (HGO) No. 23, Doctor Ignacio Morones Prieto, Monterrey 64000, Nuevo León, Mexico; 4Department of Molecular Science, U.T. Health Rio Grande Valley, McAllen, TX 78503, USA; miguel.a.garza07@utrgv.edu (M.Á.G.); genaro.ramirezcorrea@utrgv.edu (G.A.R.-C.); 5Department of Pediatrics, Division of Cardiology, Johns Hopkins University School of Medicine, Baltimore, MD 21205, USA

**Keywords:** COVID-19, SARS-CoV-2, vaccination, neutralizing antibodies, COVID-19 vaccines, ELISA

## Abstract

Neutralizing antibodies (NAs) are key immunological markers and are part of the humoral response of the adaptive immune system. NA assays determine the presence of functional antibodies to prevent SARS-CoV-2 infection. We performed a real-world evidence study to detect NAs that confer protection against SARS-CoV-2 after the application of five vaccines (Pfizer/BioNTech, AstraZeneca, Sinovac, Moderna, and CanSino) in the Mexican population. Side effects of COVID-19 vaccines and clinical and demographic factors associated with low immunogenicity were also evaluated. A total of 242 SARS-CoV-2-vaccinated subjects were recruited. Pfizer/BioNTech and Moderna proved the highest percentage of inhibition in a mono-vaccine scheme. Muscular pain, headache, and fatigue were the most common adverse events. None of the patients reported severe adverse events. We found an estimated contagion-free time of 207 (IQR: 182–231) and 187 (IQR: 184–189) days for Pfizer/BioNTech and CanSino in 12 cases in each group. On the basis of our results, we consider that the emerging vaccination strategy in Mexico is effective and safe.

## 1. Introduction

Severe acute respiratory syndrome coronavirus 2 (SARS-CoV-2) emerged in China in December 2019 and caused a worldwide severe respiratory disease (COVID-19) [1], which was declared a pandemic on 11 March 2020 by the WHO (World Health Organization) [2]. Three years after the novel coronavirus SARS-CoV-2 shook the world with a global health crisis, it remains a public health problem. COVID-19 has caused 619,161,228 cumulative infections and more than 6 million deaths worldwide as of 11 Oct 2022, according to the WHO COVID-19 Dashboard [3].

Vaccines were developed and administrated to prevent the novel coronavirus’s spread and protect at-risk populations. More than 100 vaccines have been developed, and according to the WHO, 26 COVID-19 vaccines have been evaluated in phase III clinical trials [4,5]. There are different types of COVID-19 vaccines, including the adenovirus-vectored vaccines, such as the CanSino and Oxford University/AstraZeneca COVID-19 vaccines (Ad5-CoV and ChAdOx1 nCoV-19, respectively); the mRNA vaccines from Pfizer/BioNTech and Moderna (BNT162b1 and mRNA-1273); and the inactivated vaccine from Sinovac (PiCoVacc-CoronaVac), among others [1,6,7]. To date, the Food and Drug Administration (FDA), the World Health Organization (WHO), and the European Medicine Agency (EMA) have approved four vaccines, including Pfizer/BioNTech, Moderna, and Novavax (NVX-CoV2373). Additionally, the EMA has approved Valneva, and the WHO has approved Sinopharm (BBIBP-CorV), Sinovac CoronaVac (J07BX03), Bharat Biotech Covaxin (BBV152), and CanSino.

In Mexico, the vaccination campaign started on December 2020. On the basis of the probability of infection and lethality, the government decided to start vaccination for health workers and people aged 60 years or older with or without comorbidities, followed by people a decade younger until completing the rest of the population [8,9]. Until November 2022, the regulatory agency in Mexico (the Federal Commission of Protection of Sanitary Risks, COFEPRIS) approved the following vaccines for administration in the Mexican population: Pfizer/BioNTech, AstraZeneca, Sputnik V, Sinovac, CanSino, COVAX, Moderna, Sinopharm, and Abdala [10].

Up to December 2022, Mexico had administered 225,063,079 doses of COVID vaccines (82.2% of the population with two doses), with good acceptance and proper coverage independent of the type of vaccine [4,11].

Several published studies have suggested that COVID-19 vaccines are effective and well-tolerated. However, it has been observed that the effectiveness of vaccines varies according to the sample characteristics of the population [5,12]. Effectiveness refers to how well the vaccine performs in the real world [5]. On the other hand, Mexican population data are scarce. Knowing whether and to what extent vaccine effectiveness wanes is crucial to vaccine policy decisions, such as the need for, timing, and target populations for booster doses [13].

The BNT162b2 vaccine, commercially known as the Pfizer/BioNTech COVID-19 vaccine, was the first approved by FDA and has demonstrated efficacy against the original strain of COVID-19 and other variants. It is a lipid nanoparticle-formulated, nucleoside-modified RNA (modRNA) vaccine [14] with two proline mutations to lock it in the prefusion conformation and has the ability to encode the trimerized receptor-binding domain of the spike protein of SARS-CoV-2, which allows it to mimic intact virus infection [14,15].

CanSino Biologics Inc., in collaboration with the Beijing Institute of Biotechnology, developed a vaccine that is not currently approved by the FDA; however, the vaccine received a nod from the WHO on 11 May 2022 and has been approved in over 10 countries, including China and Mexico. It is a non-replicating viral vector vaccine against SARS-CoV-2 called Adenovirus Type 5 Vector (Ad5-nCoV). Ad5-nCoV encodes the full spike protein of SARS-CoV-2 and has shown enough immunogenicity in human clinical trials [16].

Neutralizing antibody (NA) assays determine the presence of functional antibodies to prevent SARS-CoV-2 infection [17,18]. Vaccine effectiveness is measured by the quality of performance outside clinical trials [5,19]. Neutralizing antibodies are valuable tools to analyze the performance of vaccines, and they are cheap and easy to test outside clinical trials.

This real-world evidence study shows the neutralization antibody titers that confer protection against SARS-CoV-2 after the application of five vaccines (BNT162b2, AZD1222ChAdOx1, Ad5-nCoV, mRNA-1273, and CoronaVac) in the Mexican population. We also evaluated the side effects of COVID-19 vaccines and the clinical and demographic factors associated with low immunogenicity.

## 2. Materials and Methods

### 2.1. Study Population

This study was conducted following the guidelines of the Declaration of Helsinki. We collected serum samples from 242 subjects vaccinated against COVID-19 with a full vaccination scheme (143 vaccinated with Pfizer, 49 with CanSino, 21 with Sinovac, 17 with AstraZeneca, and 17 with Moderna). Vaccination schemes were considered completed if the patients had at least 2 doses of Pfizer/BioNTech, 1 dose of CanSino, 2 doses of AstraZeneca, 2 doses of Sinovac, or 2 doses of Moderna. The subjects were recruited from March to December 2021 at the Centro Universitario Contra el Cáncer (CUCC) of the Universidad Autónoma de Nuevo León (U.A.N.L.) in Monterrey, Nuevo León, México. All the subjects were ≥18 years old, signed an informed consent letter, and answered a questionnaire with clinical and demographic information (including their history of SARS-CoV-2 infection, vaccine certificates, doses, vaccine-associated side effects, etc.). Peripheral blood samples were taken to obtain serum and stored at −80 °C until analysis. The institutional ethics committee of the university hospital (Comité de Ética en Investigación del Hospital Universitario “Dr. José Eleuterio González”) approved the protocol with the registration number ON21-00028.

### 2.2. Survey and Data Collection

Patients were invited to participate in the study, and an informed consent form was signed after an interview. Clinical and demographic data were collected using a 15 min questionnaire in the SurveyMonkey^®^ platform through electronic devices (smartphones and tablets) [20].

### 2.3. Neutralizing Antibodies Detection

Blood samples were centrifuged at 3750 rpm for 10 min at room temperature to obtain serum in an Eppendorf 5804R Refrigerated Centrifuge (Hamburg, Germany). The quantification of neutralizing antibodies was performed with the cPass™ SARS-CoV-2 Neutralization Antibody Detection Kit (GenScript, Piscataway Township, NJ, USA) according to the manufacturer’s instructions. To make a semi-quantitative analysis, we added a standard curve using a monoclonal NA (MAB), SARS-CoV-2 NA, as previously described [16]. The samples and standards were read at 450 nm in a Cytation™ 3 Cell Imaging Multi-Mode Reader (BioTek^®^, Winooski, VT, USA). The following formula was used to calculate the level of signal inhibition: signal inhibition (%) = (1 − OD value of sample/OD value of negative control) × 100. The results were interpreted as follows: positive results, ≥31% of inhibition, and negative results, ≤30% of inhibition.

### 2.4. Statistical Analysis

Data analysis was performed using the IBM SPSS Statistics for Windows version 25.0 software (Armonk, NY, USA: IBM Corp.). Graphs, frequency tables, and crossed tables were constructed for categorical variables. For quantitative variables, we perform descriptive statistics, such as the mean, standard deviation, median, variance, and box plots. For comparisons of means, we used Student’s *t*-test. Data with non-parametric distributions were represented as medians with interquartile ranges (IQRs); to compare groups, we used the Mann–Whitney U-test, Kruskal–Wallis test, and Wilcoxon signed-rank test. The significance level was set at *p* < 0.05. We used the Kaplan–Meier method on Pfizer/BioNTech and CanSino for survival analysis.

## 3. Results

### 3.1. Description of Study Groups

A total of 242 SARS-CoV-2 vaccinated subjects were recruited. There were 165 females (68%), and the median age was 32 years (IQR: 25–42) (Table 1). The most frequent vaccines were Pfizer/BioNTech and CanSino, with 58% (*n* = 140) and 19% (*n* = 49), respectively (Table 2). Among comorbidities, we found overweight in 34% (*n* = 83), obesity in 24% (*n* = 59), hypertension in 9% (*n* = 22), and diabetes mellitus in 2% (*n* = 5) (Table 1).

We analyzed patients with COVID-19 before and after the vaccination scheme. There was no statistical significance in antibody titers in patients with SARS-CoV-2 infection prior to vaccination, and patients with COVID-19 after vaccination had higher antibody titers than patients without infection (*p* = 0.001) (Table 2).

### 3.2. Quantification of Neutralizing Antibodies

An exploratory analysis was carried out by the type of vaccine and their level of inhibition against SARS-CoV-2; Pfizer/BioNTech, Moderna, and CanSino showed the highest percentage of inhibition with medians of 97.23%, 97.61%, and 97.23% (IQR: 94.24–97.70, 97.23–97.94, and 73.21–97.48), respectively (Table 3). The neutralizing activity of the vaccines over time (12 months) was also measured, with a clear decrease in neutralizing antibodies after six months (Figure 1). Pfizer/BioNTech achieved better immunogenicity than CanSino (100% vs. 85%), with a significance *p* < 0.001 (Appendix A).

We found differences in immunogenicity; women had a higher percentage of inhibition than males. No statistical differences were found in any of the comorbidities analyzed.

### 3.3. Survival Analysis

We performed a survival analysis on 36 patients with COVID-19 after vaccination with Pfizer/BioNTech (*n* = 12) and CanSino (*n* = 12). The patients were RT-PCR positive, and the global survival was estimated at 187 (IQR: 184–189) days with immune protection. Pfizer/BioNTech was the most effective vaccine, with 207 (182–231) days, and CanSino had an estimation of 187 (184–189) days (Figure 2).

### 3.4. Vaccine-Associated Side Effects

The most common side effects reported were muscular pain, headache, fatigue, fever, joint pain, and shivers (Figure 3). For the first and second vaccination doses, muscular pain and headache were the most frequent side effects (52% and 41% for the first dose (*n* = 242) and 62% and 44% for the second dose (*n* = 156), respectively).

## 4. Discussion

In Mexico, more than 209 million doses have been administered, with a ratio of 162.62 total doses per 100 population [3]. Different vaccination schemes were administrated according to vaccines available, as in other developed countries; this contrasted with countries such as the United Kingdom, the United States, China, and Russia, who developed vaccines such as AstraZeneca, Pfizer/BioNTech, CanSino, and Sputnik, which were openly available to the entire population prior to other countries [21].

In Latin America (Mexico, Argentina, Chile, Ecuador, Brazil, Colombia, and Peru), the Chinese CanSino vaccine was one of the vaccines used by health authorities in the middle of the pandemic crisis, even though there was not enough information about its efficacy and security [16,22].

The combination of vaccines for generating immunity against SARS-CoV-2 has proven to be effective and safe as a vaccination strategy [23,24,25]. This allowed developing countries to complete vaccination schemes on time to reduce the cumulative incidence of COVID-19 and subsequently reinforce the vaccination scheme as the availability of vaccines increased.

In our study, Pfizer/BioNTech and Moderna showed the highest percentage of inhibition in the mono-vaccine scheme. Still, compared with an exploratory group of a heterologous vaccination scheme of CanSino (CanSino X or any other vaccine), the inhibition levels were boosted to achieve similar mono-vaccine schemes. In addition, Pfizer/BioNTech had the longest titer inhibition period, similar to that of the heterologous CanSino vaccination scheme (Appendix A) [7,16,24,26]. A higher generation of antibodies is associated with a longer duration of seroprotection [7]. According to other studies, the Pfizer/BioNTech vaccine has known global efficacy and safety with two doses of 30 μg of BNT162b2, with 91.3% (95% confidence interval (CI), 89.0 to 93.2) and specifically 96.7% (95% CI, 80.3 to 99.9) efficacy against severe and moderate symptomatic disease caused by SARS-CoV-2 infection in a period of 6 months without previous infection. Across different countries, in a wide range of age groups of both sexes with a diverse spectrum of risk factors, the efficacy was 86 to 100%. This efficacy showed a decline over time. In South Africa, for the B.1.351 variant, the efficacy is 100% (95% CI, 53.5 to 100) [14,27]. Pfizer/BioNTech showed a broad immune response with SARS-CoV-2 S-specific neutralizing antibodies for the BNT162b2 vaccine, inducing poly-specific CD4+ and CD8+ T cells for the original strain [28], with around 83.3% with neutralizing antibodies 14 days after the first dose and up to 56 days after the second dose [29].

In the phase III study for CanSino, they found that one dose of Ad5-nCoV showed a 57.5% (95% CI 39.7–70.0, *p* = 0.0026) efficacy against symptomatic PCR-confirmed COVID-19 infection 28 days or more after vaccination (21,250 participants; 45 days median duration of follow-up (IQR 36–58)). It was 63.7% efficacious against symptomatic PCR-confirmed COVID-19 infection beginning 14 days after vaccination. The Ad5-nCoV vaccine was 91.7% effective against severe disease 28 days after vaccination and 96.0% effective 14 days after vaccination. Their findings were from different countries around the world, including Mexico, Russia, and Pakistan, in participants of both sexes over 18 years old [30].

Additionally, we found a higher percentage of inhibition in women; this was previously reported in other studies, in which men had lower inhibition levels [31]. This may be explained by different interacting factors, such as environmental, genetic, and hormone factors, which differ between sexes and vary throughout life, with a general understanding that adult females mount stronger innate and adaptive immune responses than males [32].

Regarding comorbidities, including obesity, we found no statistical differences; this is important to mention because other studies have found discrepant data; some proposed a potential association between obesity and low NA titers [33], and others found that obesity had a probable booster effect in the generation of NAs [31]. This is still controversial since Mexico ranks eighth in the world for obesity; more studies should be conducted to clarify this [34].

We found an estimated median contagion-free time of 207 (IQR: 182–231), and 187 (IQR: 184–189) days for Pfizer/BioNTech and CanSino, respectively (*n* = 24). In accordance with Dr. Maria Krutik et al., who reported a seroconversion within 90–180 days, after this period, the level of NAs began to decrease [35]. The early contagion after vaccination with Pfizer/BioNTech vaccine could be explained by the fact that most of the individuals were healthcare staff on the front lines of pandemic combat who were exposed to a higher risk of contagion. CanSino was primarily administrated by personnel working in lower-risk areas of the hospital [35]. We identified seven patients (15%) vaccinated with CanSino in the single-dose modality who were negative for NAs. This agrees with another study that found similar results, with a prevalence of 11% seronegative for IgG-type antibodies [7]. Vaccine seronegative responders should be further studied to explain factors associated with poor antibody response.

Compared with other studies in which a more significant generation of neutralizing antibodies has been associated with positive COVID-19 patients before or after the vaccination scheme, our results revealed a significant generation of antibody titers in positive patients after the vaccination scheme, considering the lack of population to verify both theories with certainty as a limiting factor [36,37].

We found that muscular pain (52%), headache (41%), and fatigue (35%) were among the most common adverse events; specifically, participants who received Pfizer/BioNTech (70%) and CanSino (73%) reported at least one minor side effect. None of the vaccines had severe adverse events.

International data have shown that with two doses of Pfizer/BioNTech of 30 μg, only 27% of the patients have minimal adverse effects, characterized by short-term, mild-to-moderate pain at the injection site, fatigue, and headache within 14 days after the second dose. After six months of the vaccination, 30% had any adverse events, and only 1.2% had severe events. In adolescents, only 6% had related adverse events, and 0.6% had severe events [27,38,39].

For CanSino, in the primary safety analysis, only 0.1% of participants reported serious adverse events. In the extended safety cohort, 63·5% reported adverse events, of which headache was the most common systemic adverse event (44%). In addition, 59% reported pain at the injection site [30].

We found a higher percentage of minor side effects; this can be explained by the fact that our population was primarily young adults (median 32 years, IQR: 25–42) with similar comorbidities with an expected competent immune response. In addition, the survey allowed participants to self-report symptoms in a practical way. As expected in a real-world data study, investigator bias was not present; the results tend to be more attached to reality. We did not find any major side effects, as reported by other studies [40,41].

The main disadvantages of this study were the small sample and the vaccine diversity. Access to vaccines has been limited in Mexico. Even with this drawback, this exploratory study of five vaccine types allowed us to have an overview of neutralizing antibody titers in our population.

An advantage of this study is that we had semi-quantitative determinations to make the correlations between antibody levels and their neutralizing capacity. The blocking of neutralizing antibodies is equivalent to the gold standard neutralization test [42,43].

## 5. Conclusions

We found high NA titers in all vaccines. Pfizer/BioNTech and Moderna produced the highest antibody titers and longer immuno-protection. Pfizer/BioNTech and CanSino vaccines were tolerated and generated NAs in most participants. CanSino non-responders should be studied in greater detail to identify determinant factors, including clinical and sociodemographic characteristics involved in the low response to vaccination. The CanSino combination scheme proved to be effective and safe.

As in other developed countries, the Mexican government’s strategy attempted to apply several vaccines to reach a broad approach. This finding could be useful to define criteria for the application of vaccines in developing countries with difficult access to vaccination, such as Mexico. On the basis of our results, we consider that the emerging vaccination strategy in Mexico is effective and safe.

## Figures and Tables

**Figure 1 diagnostics-13-01194-f001:**
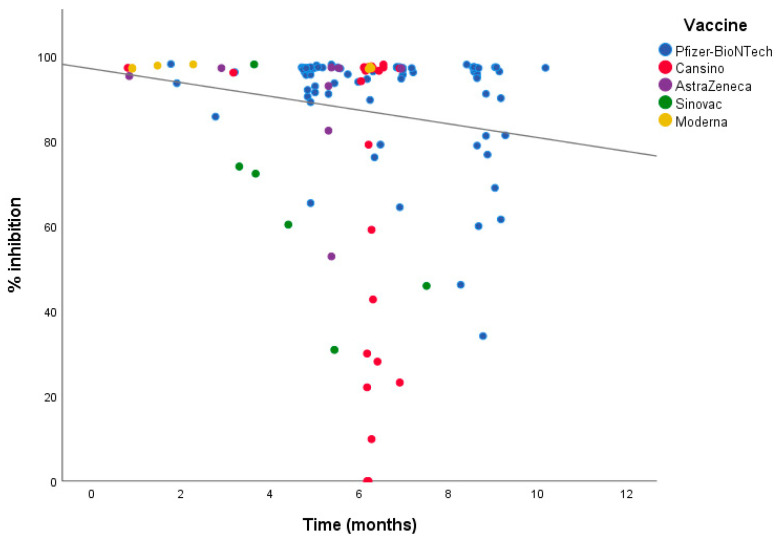
Inhibition titers of the five vaccines in a period of twelve months. Inhibition levels started to decrease at six months.

**Figure 2 diagnostics-13-01194-f002:**
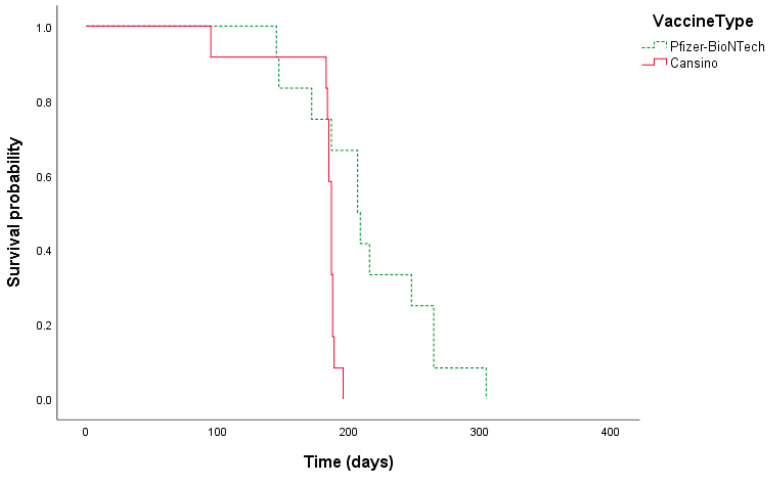
Survival analysis on COVID-19 cases after vaccination.

**Figure 3 diagnostics-13-01194-f003:**
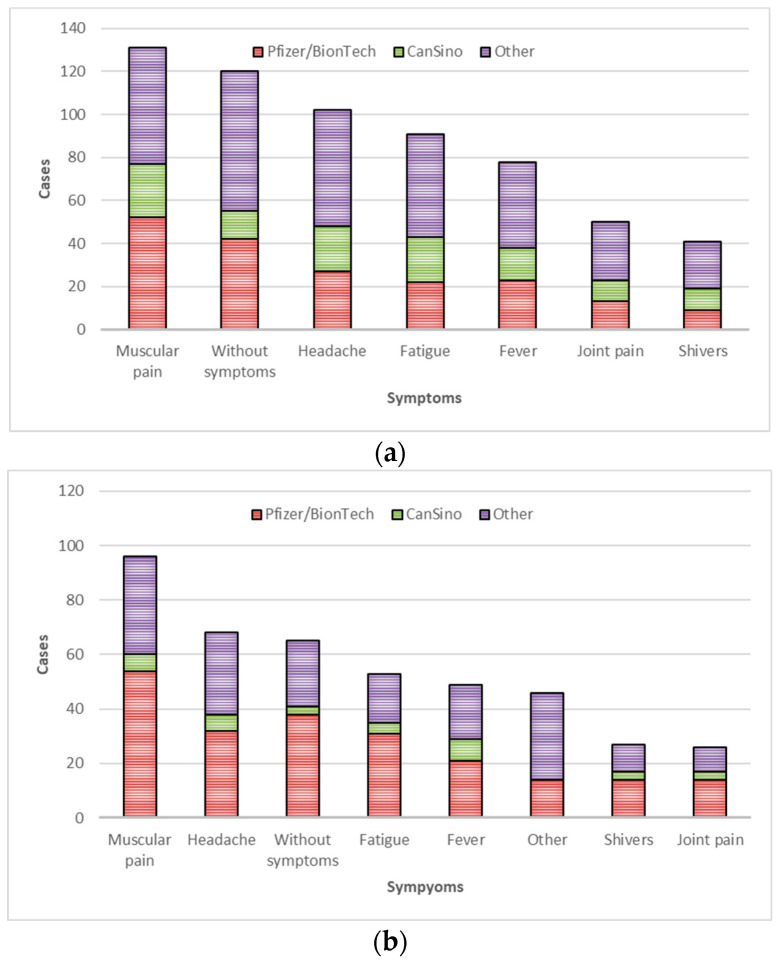
Frequency of side effects. (**a**) First dose. (**b**) Second dose.

**Table 1 diagnostics-13-01194-t001:** Clinical and demographic characteristics of the population.

Characteristics	*n*/Median
Gender, *n* (%)	
Female	165 (68%)
Male	77 (32%)
Age, median (IQR)	32 (25–42)
Occupation, *n* (%)	
Nurse/Admission	90 (37%)
Medical doctor	52 (21%)
Other	100 (42%)
BMI, median (IQR)	26.5 (23.6–29.9.16)
Overweight	83 (34%)
Obesity	59 (24%)
Physical activity, *n* (%)	127 (52%)
Income, *n* (%)	
USD 200–550	112 (46%)
>USD 550	94 (39%)
Nociceptive habits, *n* (%)	
Alcohol	160 (66%)
Tobacco	74 (30%)
Comorbidities, *n* (%)	
Hypertension	22 (9%)
Diabetes mellitus	5 (2%)
Other	45 (19%)
COVID-19 cases, *n* (%)	
Positive before vaccine	75 (31%)
Positive after vaccine	33 (14%)
Characteristics	*n*/median
Gender, *n* (%)	
Female	165 (68%)
Male	77 (32%)
Age, median (IQR)	32 (25–42)
Occupation, *n* (%)	
Nurse/Admission	90 (37%)
Medical doctor	52 (21%)
BMI, median (IQR)	26.5 (23.6–29.9.16)
Overweight	83 (34%)
Obesity	59 (24%)
Physical activity, *n* (%)	127 (52%)
Income, *n* (%)	
USD 200–550	112 (46%)
>USD 550	94 (39%)
Nociceptive habits, *n* (%)	
Alcohol	160 (66%)
Tobacco	74 (30%)
Comorbidities, *n* (%)	
Hypertension	22 (9%)
Diabetes mellitus	5 (2%)
Other	45 (19%)
COVID-19 background, *n* (%)	
Positive before vaccine	75 (31%)
Positive after vaccine	33 (14%)

IQR—interquartile range.

**Table 2 diagnostics-13-01194-t002:** Vaccine distribution.

Vaccines	*n* (%)
Pfizer/BioNTech	140 (58%)
CanSino	49 (19%)
AstraZeneca	17 (7%)
Sinovac	21 (8%)
Moderna	15 (6%)

**Table 3 diagnostics-13-01194-t003:** Vaccine inhibition titers.

	Pfizer/BioNTech		CanSino	AstraZeneca	Sinovac	Moderna
	*n*	143	49	17	21	17
IQR						
25		94.24	73.21	78.19	62.39	97.23
median		97.23	97.23	97.18	74.05	97.61
75		97.70	97.48	97.92	97.87	97.94

## Data Availability

Not applicable.

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
