# Peer review of "SARS-CoV-2 Neutralizing Antibodies in Mexican Population: A Five Vaccine Comparison"

_diagnostics, 2023, doi:10.3390/diagnostics13061194_

Round 1
Reviewer 1 Report
The study by Alcorta-Nuñez et al evaluated vaccine-induced antibody responses measured by a surrogate neutralization assay. The study cohort is heterogenous with regards to the vaccine type received (57% Pfizer/BioNTech [mRNA], 19.5% CanSino [Adv-vectored], 8% Sinovac [inactivated], 6.8% AstraZeneca [Adv-vectored], 6.8% Moderna [mRNA, and 1.6% Janssen [Adv-vectored]). Some limitations of the study revolve around factors that may confound antibody responses that were not considered or adjusted for in the study (please see below).
Major comments
There are several factors that modulate antibody responses after SARS-CoV-2 vaccination that include, but are not limited to i) age, ii) SARS-CoV-2 infection before or after vaccination, iii) vaccine modality, iv) heterologous prime-boost regimens, v) sampling time after vaccination, and vi) time between vaccinations.
SARS-CoV-2 infection before or after vaccination. A single vaccination following a previous infection induces strong antibody recall responses to a similar level induced by a 3rd dose after a primary 2-dose regimen, which is not boosted by a second vaccination (DOI: 10.1056/NEJMc2101667; DOI:https://doi.org/10.1016/j.ebiom.2021.103656). Infection after vaccination also increases antibody responses. Therefore, SARS-CoV-2 infections may confound the levels of surrogate neutralization levels measured in the study. The authors report that 31% of vaccinees had a SARS-CoV-2 infection before vaccination and 14% after vaccination. If SARS-CoV-2 infections were overrepresented in any of the vaccine groups and considering the low sample sizes, one may overestimated the vaccine-induced antibody responses in this study.
Vaccine modality. SARS-CoV-2 mRNA vaccines induce higher antibody responses compared to SARS-CoV-2 adenoviral vectored vaccines. The authors compare antibody responses between the vaccine modalities, although the sample sizes for some are quite small and the demographics of each group is not presented or number of SARS-CoV-2 infections. This makes it difficult to compare the antibody responses for the cohort studied.
Heterologous prime-boost. The authors cite a papers (ref 20-22) about the benefits of heterologous prime-boost regimens and mention that there are individuals in their study cohort who received CanSino in combination with another vaccine (CanSino combined scheme) in the survival analysis and in the discussion. However, it is not presented as a separate group in the materials and methods or initial cohort description or antibody response analysis. How was combination vaccination considered in the antibody analysis?
Sampling time after vaccination. From Figure 1, it would appear as though the serum samples were primarily taken more than 4 months after vaccination. Since antibody responses may wane at different rates for different vaccine modalities and different vaccine modalities in general give different antibody levels (eg. mRNA induces higher antibody levels compared to Adv-vectored vaccines), it becomes very difficult to conclude waning immunity by combining all vaccine types in a single analysis. Ideally, each vaccine should be presented on its own. Was there a difference in timing of sample collection post-vaccination for the different vaccine types?
Specific comments:
1) Line 66. Please add references
2) Line 72-73. If references is made to vaccine effectiveness in a paragraph centered around neutralizing antibodies, it will be helpful to describe the relationship between neutralizing antibodies and vaccine protection.
3) Line 84-85. Please include percentages
4) Line 108. Please include the principle of this surrogate neutralization assay. That it, it specifically measures the ability of antibodies to block the interaction between the SARS-CoV-2 spike receptor binding domain and the ACE-2 receptor.
5) Line 108. Were samples tested in replicate?
6) Relating to the data presented in Table 1, please describe in the materials and methods how each characteristic was defined. For example, assuming overweight and obesity was defined by body mass index (BMI), describe the BMI intervals for these categories; income – was it weekly, monthly, yearly; nociceptive habits – number of units per day; comorbidities – what are the other comorbidities; etc.
7) Relating to table 2, please describe in the manuscript how many doses are considered completed primary vaccination for the different vaccine types e.g. 2 doses for Pfizer/BioNTech and Moderna, 1 dose for Janssen, etc. Include in this table the proportion of individuals that received mono-vaccine regimens and those who received heterologous vaccine regimens.
8) Line 150. ‘Efficacy’ refers to vaccine protection against infection, which was not measured in Figure 1.
9) Table 3. If possible, please present the individual data points in graphs. It will also be helpful to place data for the same vaccine modalities next to each other.
10) Line 163. Survival analysis. The sample sizes are really small and any other confounders in the three groups are not presented in the manuscript, e.g. age, gender, weight, comorbidities. This makes it difficult to conclude that one vaccine type is superior to another.
11) Line 211. Please clarify that this refers to heterologous vaccination, the consecutive vaccination of an individual with two different types of vaccines.
12) Line 255-256. Safety for CanSino was not evaluated in this study.
Author Response
Major comments
There are several factors that modulate antibody responses after SARS-CoV-2 vaccination that include, but are not limited to i) age, ii) SARS-CoV-2 infection before or after vaccination, iii) vaccine modality, iv) heterologous prime-boost regimens, v) sampling time after vaccination, and vi) time between vaccinations.
Comment 1: SARS-CoV-2 infection before or after vaccination. A single vaccination following a previous infection induces strong antibody recall responses to a similar level induced by a 3rd dose after a primary 2-dose regimen, which is not boosted by a second vaccination (DOI: 10.1056/NEJMc2101667; DOI:https://doi.org/10.1016/j.ebiom.2021.103656). Infection after vaccination also increases antibody responses. Therefore, SARS-CoV-2 infections may confound the levels of surrogate neutralization levels measured in the study. The authors report that 31% of vaccinees had a SARS-CoV-2 infection before vaccination and 14% after vaccination. If SARS-CoV-2 infections were overrepresented in any of the vaccine groups and considering the low sample sizes, one may overestimated the vaccine-induced antibody responses in this study.
Answer: We made a Mann-Whitney U Test for non-parametric variables. We analyzed patients with COVID-19 before and after the vaccination scheme. There was no statistical significance in antibody titers in patients with SARS-CoV-2 infection prior to vaccination, and patients with COVID-19 post-vaccination had higher antibody titers versus patients without infection (p= 0.001) (Table 2).
We added a paragraph in the discussion section; we also added the references you recommended to us: Compared with other studies where a more significant generation of neutralizing antibodies has been associated with positive COVID-19 patients before or after the vaccination scheme, our results prove a significant generation of antibody titers in positive patients after the vaccination scheme, considering the lack of population to verify both theories with certainty as limiting factor [35,36].
Comment 2: Vaccine modality. SARS-CoV-2 mRNA vaccines induce higher antibody responses compared to SARS-CoV-2 adenoviral vectored vaccines. The authors compare antibody responses between the vaccine modalities, although the sample sizes for some are quite small and the demographics of each group is not presented or number of SARS-CoV-2 infections. This makes it difficult to compare the antibody responses for the cohort studied.
Answer: This is because, in Mexico, health workers were vaccinated mainly with the Pfizer and Moderna vaccines (https://doi.org/10.3855/jidc.15545, doi: 10.3389/fpubh.2022.834744). We decided to exclude the Jansen vaccination group because of the small sample. Knotted to this Johnson and Johnson vaccine had a low distribution in Mexico.
We added a paragraph in the discussion section: “The main disadvantages of this study were the small sample and the vaccine diversity. Access to vaccines has been limited in Mexico. Even with this drawback, this exploratory study in five vaccine types allowed us to have an overview of neutralizing antibody titers in our population”.
We changed the title of the article to better describe our study as Follows: SARS-CoV-2 neutralizing antibodies in Mexican population: a five-vaccine comparison.
Comment 3: Heterologous prime-boost. The authors cite a paper (ref 20-22) about the benefits of heterologous prime-boost regimens and mention that there are individuals in their study cohort who received CanSino in combination with another vaccine (CanSino combined scheme) in the survival analysis and in the discussion. However, it is not presented as a separate group in the materials and methods or initial cohort description or antibody response analysis. How was combination vaccination considered in the antibody analysis?
Answer: We decided to eliminate the heterologous vaccine scheme because of the small sample.
We explained it in the Discussion section as follows: In our study, Pfizer/BioNTech and Moderna proved the highest percentage of inhibition in the mono-vaccine scheme. Still, compared with an exploratory group of a heterologous vaccination scheme of CanSino (CanSino X or any other vaccine), the inhibition levels were boosted to achieve similar mono-vaccine schemes. Also, Pfizer/BioNTech had the most extended titer inhibition period, like the heterologous vaccination CanSino scheme (Table S1) [7,16,24,26].
Comment 4: Sampling time after vaccination. From Figure 1, it would appear as though the serum samples were primarily taken more than 4 months after vaccination. Since antibody responses may wane at different rates for different vaccine modalities and different vaccine modalities in general give different antibody levels (eg. mRNA induces higher antibody levels compared to Adv-vectored vaccines), it becomes very difficult to conclude waning immunity by combining all vaccine types in a single analysis. Ideally, each vaccine should be presented on its own. Was there a difference in timing of sample collection post-vaccination for the different vaccine types?
Answer: There was no homogeneity in the timing collection, but we had the registry of vaccination of all participants so we could manage to know the correlation of titters in time (Figure 1).
Specific comments:
Comment 5: Line 66. Please add references.
Answer: In line 66: However, it has been observed that the effectiveness of vaccines varies according to the sample characteristics of the population [5,12].
We reference the following papers:
5.- Fiolet, T.; Kherabi, Y.; MacDonald, C.J.; Ghosn, J.; Peiffer-Smadja, N. Comparing COVID-19 vaccines for their characteristics, efficacy and effectiveness against SARS-CoV-2 and variants of concern: a narrative review. Clin Microbiol Infect 2022, 28, 202-221, doi:10.1016/j.cmi.2021.10.005.
12.- He, X.; Su, J.; Ma, Y.; Zhang, W.; Tang, S. A comprehensive analysis of the efficacy and effectiveness of COVID-19 vaccines. Front Immunol 2022, 13, 945930, doi:10.3389/fimmu.2022.945930.
Comment 6: Line 72-73. If references is made to vaccine effectiveness in a paragraph centered around neutralizing antibodies, it will be helpful to describe the relationship between neutralizing antibodies and vaccine protection.
Answer: We added a new reference a modified the text as follows:
Effectiveness refers to how well the vaccine performs in the real world; neutralizing antibodies are generated within weeks after immunization or infection and can provide evidence of protective immunity [5,13]. Neutralizing antibodies have been important in vaccine development and determining seroprevalence to assist the government in adjusting policy decisions [13].
Reference 13: https://doi.org/10.3390/v14071560
Comment 7: Line 84-85. Please include percentages
Answer: We added the percentages as follows:
This study was conducted following the Declaration of Helsinki. We collected serum samples from 247 subjects vaccinated against COVID-19 with a full vaccination scheme. (143 (57.89%) were vaccinated with Pfizer, 49 (19.83%) with CanSino, 21 (8.59%) with Sinovac, 17 (6.88%) with AstraZeneca, and 17 (6.88%) with Moderna.
We delete the Jansen vaccine group because of the small sample.
Comment 8: Line 108. Please include the principle of this surrogate neutralization assay. That it, it specifically measures the ability of antibodies to block the interaction between the SARS-CoV-2 spike receptor binding domain and the ACE-2 receptor.
Answer: The kit detects and measures circulating neutralizing antibodies against the SARS-CoV-2 virus. Neutralizing antibodies generated by the immune system after COVID-19 infection or vaccination function by blocking the interaction between the receptor binding domain (RBD) of the SARS-CoV-2 spike glycoprotein and the ACE2 human cell surface receptor (https://www.genscript.com/covid-19-detection-cpass.html).
Expectedly, assays targeting SARS-CoV-2 S protein, particularly the RBD, showed the best correlation with neutralizing activity, indicating that these assays could serve as reliable and high-throughput assays for predicting the presence of nAbs (DOI: 10.1186/s40779-021-00342-3). Further, GenScript cPass sVNT exhibited the best linear correlation with pVNT in samples collected from both infected and vaccinated individuals. These results are consistent with previous studies assessing the performance of GenScript sVNT compared to MNA and pVNT, demonstrating high specificity and even more sensitivity (https://doi.org/10.1038/s41598-022-21317-x).
We modified the Material and Methods section as follows:
“Blood samples were centrifuged at 3750 rpm for 10 minutes at room temperature to obtain serum in an Eppendorf 5804R Refrigerated Centrifuge (Hamburg, Germany). The quantification of neutralizing antibodies was performed with the cPass™ SARS-CoV-2 Neutralization Antibody Detection Kit (GenScript, Piscataway Township, NJ, USA), according to the manufacturer's instructions. Neutralizing antibodies generated by the immune system after COVID-19 infection or vaccination function by blocking the interaction between the receptor binding domain (RBD) of the SARS-CoV-2 spike glycoprotein and the ACE2 human cell surface receptor. To make a semi-quantitative analysis, we added a standard curve using the monoclonal NA (MAB), SARS-CoV-2 NA, as previously described [16]. The samples and standards were read at 450 nm in a Cytation™ 3 Cell Imaging Multi-Mode Reader (BioTek®, Winooski, VT, USA). The following formula was used to calculate the level of signal inhibition: signal inhibition (%) = (1- OD value of sample/OD value of negative control) x 100. The results were interpreted as follows: positive results ≥31% and negative ≤30% of inhibition.”
Comment 9: Line 108. Were samples tested in replicate?
Answer: We tested only a single replicate per sample.
Comment 10: Relating to the data presented in Table 1, please describe in the materials and methods how each characteristic was defined. For example, assuming overweight and obesity was defined by body mass index (BMI), describe the BMI intervals for these categories; income – was it weekly, monthly, yearly; nociceptive habits – number of units per day; comorbidities – what are the other comorbidities; etc.
Answer: We added the information requested in table 1. BMI was measured quantitatively; alcohol and tobacco were measured in a qualitative fashion (yes or no question). We also explained that the income was monthly.
Comment 11: Relating to table 2, please describe in the manuscript how many doses are considered completed primary vaccination for the different vaccine types e.g. 2 doses for Pfizer/BioNTech and Moderna, 1 dose for Janssen, etc. Include in this table the proportion of individuals that received mono-vaccine regimens and those who received heterologous vaccine regimens.
Answer: Table 2 is now referenced in line 198, and the definition of a completed vaccination scheme was explained in section 2.1 of material and methods in lines 138-142.
This study was conducted following the Declaration of Helsinki. We collected serum samples from 247 subjects vaccinated against COVID-19 with a full vaccination scheme: 143 (57.89%) vaccinated with Pfizer, 49 (19.83%) with CanSino, 21 (8.59%) with Sinovac, 17 (6.88%) with AstraZeneca, and 17 (6.88%) with Moderna. Vaccination schemes were considered completed depending on the vaccine as follows: at least 2 doses with Pfizer/BioNTech, 1 dose of CanSino, 2 doses of AstraZeneca, 2 doses of Sinovac, and 2 doses for Moderna.
Comment 12: Line 150. ‘Efficacy’ refers to vaccine protection against infection, which was not measured in Figure 1.
Answer: Efficacy was changed to neutralizing activity in line 224.
Comment 13: Table 3. If possible, please present the individual data points in graphs. It will also be helpful to place data for the same vaccine modalities next to each other.
Answer: We understand what you're saying, but we thought that Table 3 presented the results clearly. However, we made the graphic, and we are open to the editor's suggestions.
Comment 14: Line 163. Survival analysis. The sample sizes are really small and any other confounders in the three groups are not presented in the manuscript, e.g. age, gender, weight, comorbidities. This makes it difficult to conclude that one vaccine type is superior to another.
Answer: We decided to change the survival analysis and include only Pfizer and CanSino (Figure 2); we excluded heterologous vaccination because of the small n of the group. We also added this topic in the discussion section.
We considered that we have a homogeneous sample, considering that age was from 27 to 42 years, with a mean of 33 years, with similar comorbidities. Other confounding factors were also discussed.
Additionally, we found a higher percentage of inhibition in women; this has been previously reported in other studies where men had lower inhibition levels [31]. This may be explained by different interacting factors such as environmental, genetic, and hormone that differ in both sexes that varies throughout life, with a general understanding that adult females mount stronger innate and adaptive immune responses than males [32].
Regarding comorbidities, we found no statistical difference, including obesity; this is important to mention because other studies have found discrepant data, some proposing a potential association between obesity and low NA titers [33], and others found obesity as a probable booster effect in the generation of NA [31]. This is still controversial since Mexico has the 8° world rank of obesity; more studies should be conducted to clarify this [34].
Comment 15: Line 211. Please clarify that this refers to heterologous vaccination, the consecutive vaccination of an individual with two different types of vaccines.
Answer: All the time combined scheme mentioned in the manuscript was changed to heterologous vaccination.
Comment 16: Line 255-256. Safety for CanSino was not evaluated in this study.
Answer: There was confusion in the submission of the paper, and e PDF is different from the word data. In the word archive, CanSino safety was evaluated as a single dose and as heterologous vaccination with either Pfizer or Moderna, as seen in lines 261-263, and the description for figure 3 was modified to better reflect this.

Reviewer 2 Report
The author shows that neutralizing antibody titers can provide protection against SARS-CoV-2 after the application of six vaccines (BNT162b2, AZD1222ChAdOx1, Ad5-nCoV, mRNA-1273, CoronaVac and Ad26. COV2. S) in the Mexican population. They also evaluated the side effects of the COVID-19 vaccine and the clinical and demographic factors associated with low immunogenicity. I have a number of comments and questions regarding the manuscript as listed below:
1. The evaluation of the effect of six different vaccines may not be of great significance to medical workers. Medical workers have different protection requirements, and the time of infection is not closely related to vaccination. The authors should expand and include different groups.
2. The background of COVID-19 in the included population is different. Some people have been infected with COVID-19, so neutralizing antibodies have been produced. Re vaccination may be a double protection. The authors should group people with different backgrounds of COVID-19 into 6 different vaccines, and then make specific analysis.
I believe that the authors could address these comments but they would need to conduct additional experiments and I do not conceive how this can be done in a suitable time frame.
Author Response
The author shows that neutralizing antibody titers can provide protection against SARS-CoV-2 after the application of six vaccines (BNT162b2, AZD1222ChAdOx1, Ad5-nCoV, mRNA-1273, CoronaVac and Ad26. COV2. S) in the Mexican population. They also evaluated the side effects of the COVID-19 vaccine and the clinical and demographic factors associated with low immunogenicity. I have a number of comments and questions regarding the manuscript as listed below:
Comment 1: The evaluation of the effect of six different vaccines may not be of great significance to medical workers. Medical workers have different protection requirements, and the time of infection is not closely related to vaccination. The authors should expand and include different groups.
Answer: Not all analyzed individuals were health providers. There were also general population and teachers, corresponding to 22% of the total population 242/55. When we made the analysis, we found no difference between healthcare staff and the general population. This is reflected in section 3 of the results in “Table 1 Clinical and demographic characteristics of the population.”.
Comment 2: The background of COVID-19 in the included population is different. Some people have been infected with COVID-19, so neutralizing antibodies have been produced. Re vaccination may be a double protection. The authors should group people with different backgrounds of COVID-19 into 6 different vaccines, and then make specific analysis. I believe that the authors could address these comments but they would need to conduct additional experiments and I do not conceive how this can be done in a suitable time frame.
Answer: We made a Mann-Whitney U Test for non-parametric variables. We analyzed patients with COVID-19 before and after the vaccination scheme. There was no statistical significance in antibody titers in patients with SARS-CoV-2 infection prior to vaccination, and patients with COVID-19 post-vaccination had higher antibody titers versus patients without infection (p= 0.001) (Table 2).
We added a paragraph in the discussion section; we also added the references you recommended to us: Compared with other studies where a more significant generation of neutralizing antibodies has been associated with positive COVID-19 patients before or after the vaccination scheme, our results prove a significant generation of antibody titers in positive patients after the vaccination scheme, considering the lack of population to verify both theories with certainty as limiting factor [35,36].

Reviewer 3 Report
I have a comment about Johnson vaccines ? The cause of low number only 4 patients vaccinated? Why
Author Response
Comment 1: I have a comment about Johnson vaccines? The cause of low number only 4 patients vaccinated? Why (justificar con lo que nos paso Diana (véase arriba) expander en discusión) (DCPI)
Answer: This is because health workers were vaccinated mainly with the Pfizer and Moderna vaccines (https://doi.org/10.3855/jidc.15545, doi: 10.3389/fpubh.2022.834744). Knotted to this Johnson and Johnson vaccine had a low distribution in Mexico. On May 27, 2021, the government of Mexico received, as part of a donation by the government of President Biden, 1.35 million vaccines from Johnson and Johnson. These vaccines were used for inhabitants of the border of Mexico with the United States as part of a strategy of both countries to open them (https://www.gob.mx/sre/prensa/bilateral-agreement-with-us-leads-to-the-arrival-of-1-35-million-doses-of-the-johnson-johnson-vaccine, https://www.insumosparasalud.com/ipsnews2/vaccines-in-mexico, https://www.elpasotimes.com/story/news/2021/06/29/mexico-sends-j-j-covid-vaccine-tijuana-none-juarez/7796123002/).

Reviewer 4 Report
Dear Authors,
In my opinion it would be useful to set up a table in which to includethe various information on the different vaccines reported in the rows from 46 to 52. What was the selection criterion for administering the various
types of vaccines to the various subjects? There is a reason to justify the fact that the most administered vaccines
were Pfizer/BioNTech and CanSino?
Author Response
Comment 1: In my opinion it would be useful to set up a table in which to include the various information on the different vaccines reported in the rows from 46 to 52. What was the selection criterion for administering the various types of vaccines to the various subjects? There is a reason to justify the fact that the most administered vaccines were Pfizer/BioNTech and CanSino?
Answer: We recruited subjects with a complete vaccination scheme.
The availability of vaccines in Mexico depends on commercial availability and that they had approval from the regulatory agency in Mexico COFEPRIS, Pfizer was the first vaccine to obtain a permit (December 11, 2020), followed by Astra Zeneca (January 4, 2021), Sputnik (February 2, 2021) and so on for distribution in our country. The population selection criteria were based on vaccinating the people most susceptible to developing complications from COVID-19 (over 60 years of age or with comorbidity); this is to reduce the number of deaths and complications related to the disease. Due to the vaccination backlog in our country, especially for health personnel (who are most of our study population), many of these went to be vaccinated at the border, which is why the variety of vaccine types in the study subjects. (https://coronavirus.gob.mx/wp-content/uploads/2021/04/28Abr2021_13h00_PNVx_COVID_19.pdf, https://coronavirus.gob.mx/wp-content/uploads/2021/10/2021.10.04_CP_Salud_CTD_COVID-19.pdf, https://www.wilsoncenter.org/article/infographic-mexicos-vaccine-supply-and-distribution-efforts
https://coronavirus.gob.mx/wp-content/uploads/2021/04/28Abr2021_13h00_PNVx_COVID_19.pdf
https://coronavirus.gob.mx/wp-content/uploads/2021/10/2021.10.04_CP_Salud_CTD_COVID-19.pdf
https://www.wilsoncenter.org/article/infographic-mexicos-vaccine-supply-and-distribution-efforts

Reviewer 5 Report
Alcorta et al have compared six vaccines in dealing with Covid-19 for a certain population in Mexico to show NA confers protection against SARS-CoV-2 after the application of six vaccines. Even Though such analysis is essential for vaccine development but it doesn't study a large population, so you can't rely on its results and claims. Considering this shortcoming, I would not recommend publishing.
Author Response
Comment 1: Alcorta et al have compared six vaccines in dealing with Covid-19 for a certain population in Mexico to show NA confers protection against SARS-CoV-2 after the application of six vaccines. Even Though such analysis is essential for vaccine development, but it doesn't study a large population, so you can't rely on its results and claims. Considering this shortcoming, I would not recommend publishing.
Answer: We attended all the suggestions of the rest of the reviewers and improve the manuscript. We included the limitations of the study in the discussion section. We also changed the statistical analysis to improve our results.

Round 2
Reviewer 1 Report
The authors addressed all comments to satisfaction.
Reviewer 2 Report
After modification, it can meet the publishing requirements.
Reviewer 5 Report
Agree